# Chromatographic Analysis and Enzyme Inhibition Potential of *Reynoutria japonica* Houtt.: Computational Docking, ADME, Pharmacokinetic, and Toxicokinetic Analyses of the Major Compounds

**DOI:** 10.3390/ph18030408

**Published:** 2025-03-14

**Authors:** Tugsen Buyukyildirim, Fatma Sezer Senol Deniz, Osman Tugay, Ramin Ekhteiari Salmas, Onur Kenan Ulutas, Ibrahim Ayhan Aysal, Ilkay Erdogan Orhan

**Affiliations:** 1Department of Pharmacognosy, Faculty of Pharmacy, Selcuk University, 42071 Konya, Türkiye; tugsen.dogru@selcuk.edu.tr; 2Department of Pharmacognosy, Faculty of Pharmacy, Gazi University, 06330 Ankara, Türkiye; fssenol@gazi.edu.tr; 3Department of Pharmaceutical Botany, Faculty of Pharmacy, Selcuk University, 42071 Konya, Türkiye; otugay@selcuk.edu.tr; 4Department of Chemistry, Britannia House, King’s College, London WC2R 2LS, UK; ramin.ekhteiari@gmail.com; 5Department of Pharmaceutical Toxicology, Faculty of Pharmacy, Gazi University, 06330 Ankara, Türkiye; onurkenan@gazi.edu.tr; 6Department of Analytical Chemistry, Faculty of Pharmacy, Gazi University, 06330 Ankara, Türkiye; ayhanaysal@outlook.com; 7Department of Pharmacognosy, Faculty of Pharmacy, Lokman Hekim University, 06510 Ankara, Türkiye; 8Principal Member of Turkish Academy of Sciences (TÜBA), Vedat Dalokay Street, No. 112, 06670 Ankara, Türkiye

**Keywords:** *Reynoutria japonica*, resveratrol, piceid, LC-MS QTOF, enzyme inhibition, in silico ADME, in silico toxicokinetic, docking simulations

## Abstract

**Background:** *Reynoutria japonica* Houtt. has been used for inflammatory diseases, skin burns, and high cholesterol in traditional Chinese medicine, and the roots and rhizomes of the plant were registered in the Chinese Pharmacopoeia. This study evaluated the enzyme inhibitory activities of *R. japonica* extracts from Türkiye. Its major phytochemical content was elucidated, molecular interaction studies of the main compounds were conducted, and toxicokinetic predictions and absorption, distribution, metabolism, and elimination studies were performed with in silico methods. **Methods**: *R. japonica* extracts were tested for their enzyme inhibitory activities using an ELISA microplate reader. The phytochemical profile was elucidated by LC-MS QTOF. Docking and other in silico studies evaluated interactions of its main components with cholinesterase, collagenase, and elastase. **Results**: *R. japonica* exhibited significant cholinesterase inhibitory effectiveness, while the stem and root extracts showed moderate tyrosinase inhibition. *R. japonica* leaf (IC_50_ = 117.20 ± 4.84 g/mL) and flower extracts (IC_50_ = 111.40 ± 1.45 µg/mL) exhibited considerable elastase activity. *R. japonica* leaf (IC_50_ = 171.00 ± 6.76 g/mL) and root (IC_50_ = 160.00 ± 6.81 g/mL) extracts displayed similar and potent collagenase inhibition. In the LC-MS QTOF analysis, procyanidin dimer, catechin, piceid, torachrysone, and its glucoside isomers were identified as the major components and resveratrol as the minor component. Galloylglucose showed the strongest binding at cholinesterase *via* key hydrogen bonds, while emodin-6-glucoside and emodin formed stable interactions with elastase. Piceid displayed significant polar and water-mediated contacts with collagenase. These findings underscore the potential of these ligands as protein inhibitors. In silico predictions reveal that emodin possessed the most favorable drug-like properties but posed potential interaction risks. **Conclusions**: This research represents the first investigation of the bioactivity and phytochemistry of *R. japonica* grown and documented in 2020 in Türkiye. Our findings point out that *R. japonica* could be used for cosmetic purposes, and further studies on neurological disorders could be performed.

## 1. Introduction

*Reynoutria japonica* Houtt. [syn: *Polygonum cuspidatum* Sieb.&Zucc., *Fallopia japonica* (Houtt.) Ronse Decr.] (Polygonaceae), known as Japanese knotweed in English, is an East Asian natural herbaceous perennial geophyte that grows as an early successional species in lava and gravel fields [1]. The Pharmacopoeia of the People’s Republic of China has listed *R. japonica* since 1977, and emodin and polydatin are employed as marker compounds to describe its pharmaceutical quality [2]. *R. japonica* is known to be a major source of natural resveratrol sold in dietary supplement markets worldwide [3]. The presence of various secondary metabolite groups such as stilbenes (resveratrol and its derivatives), procyanidins, flavonoids (quercetin, kaempferol, and luteolin glycosides), phenolic acids, anthraquinones (emodin derivatives), phenylpropanoids, lignin oligomers, and naphthalenes was reported in the plant [4,5]. Due to its rich phytochemistry, roots, rhizomes, and aerial parts (stems, leaves, and flowers) of *R. japonica* have been demonstrated to exhibit antibacterial activity, antioxidant effects, anticancer, antiproliferative, apoptotic properties, gastroprotective, neuroprotection effect, anti-inflammatory, and antiviral activity [6,7,8]. *R. japonica* was also published in 2020 as a new botanical record for the flora of Türkiye, and there have been no studies up to date with this plant from Türkiye in the literature [9].

Resveratrol (3,5,4′-trihydroxy-*trans*-stilbene) is a naturally occurring reputed polyphenol with a molecular weight of 228.2 g/mol [10]. Resveratrol has a planar structure that gives it hydrophobic properties. Therefore, it has high-affinity interactions with the hydrophobic domains of target protein molecules. Also, three polar hydroxy groups participate in hydrogen bonding with amino acid side chains of the targeted proteins. For this reason, it interacts with many proteins and has broad pharmacological effects [11]. Over 244 ongoing clinical studies have been conducted on resveratrol’s efficacy, safety, and pharmacokinetics [12]. A vast amount of research has been carried out on resveratrol to be evaluated for being a drug candidate molecule or food supplement against various cancer types (such as prostate cancer, colorectal cancer, breast cancer), Alzheimer’s disease (AD), ischemic stroke, coronary artery, atherosclerosis, hypertension, inflammation, and oxidative stress are available [13].

AD is a progressive neurological condition that typically affects older people and reduces a person’s memory and cognitive ability [14]. In AD pathology, the lack of the neurotransmitter acetylcholine, known as the cholinergic hypothesis, has brought acetyl/butyrylcholinesterase (AChE/BChE) inhibitors to the fore in the treatment. Hence, the cholinergic hypothesis has been one of the most studied and influential hypotheses, resulting in three clinical drugs (donepezil, galantamine, and rivastigmine) approved for treating symptoms of AD [15]. However, due to the side effects and limited efficacy of the current drugs, developing new AChE/BChE inhibitors is still essential to increase alternative treatment modalities in AD.

Aging and ultraviolet radiation increase collagenase and elastase activity and cause the breakdown of collagen fibers and elastin. These changes cause wrinkles and sagging on the skin [16]. The inhibition of these enzymes has attracted attention recently due to the positive effects of their inhibitors in preventing skin aging and connective tissue diseases [17]. Tyrosinase (TYR) is a copper-bound enzyme that forms an essential step in the biosynthesis of the dark macromolecule melanin [18]. TYR inhibitors are used in various fields, such as spot lightening, skin lightening, and melanogenesis treatment [19].

The medicinal use of *R. japonica* dates back to the 1800s. The plant, which contains different groups of secondary metabolites, is one of the few higher plant species that contains resveratrol and its derivatives, known as phytoalexin [20]. The inhibitory activities of AChE and BChE, which are important for treating AD, and on collagenase, elastase, and TYR as cosmetic-related enzymes of *R. japonica* extracts were evaluated. In addition, the phytochemical profile of *R. japonica* was elucidated by LC-MS QTOF, and the resveratrol contents of this species collected from Türkiye were determined. In silico studies provide valuable guidance for future drug development and optimization strategies, particularly in choosing appropriate delivery systems and considering potential modifications to improve specific properties while maintaining therapeutic efficacy. Thus, molecular interaction simulations, along with toxicokinetic predictions and absorption, distribution, metabolism, and elimination (ADME) studies on the main compounds were carried out using in silico methods.

## 2. Results

### 2.1. Enzyme Inhibition Results

All tested extracts of *R. japonica* exhibited notable inhibitory activities against AChE. While *R. japonica* stem (IC_50_ = 25.06 ± 0.52 µg/mL) extract displayed the highest AChE inhibition, leaf (IC_50_ = 53.31 ± 3.38 µg/mL), flower (IC_50_ = 64.44 ± 4.69 µg/mL) and root (IC_50_ = 120.10 ± 0.14 µg/mL) extracts disclosed marked inhibitory effect. On the other hand, *R. japonica* stem extract (IC_50_ = 159.78 ± 3.50 µg/mL) caused the highest BChE inhibition, whereas moderate inhibition was caused by the leaf (IC_50_ = 202.50 ± 1.41 µg/mL), flower (41.13%), and root (20.15%) extracts (Table 1).

While stem (26.03%) and root (19.46%) extracts of *R. japonica* showed moderate TYR enzyme inhibition, no effect was observed in *R. japonica* flower and leaf extracts. *R. japonica* flower (IC_50_ = 111.40 ± 1.45 µg/mL) and leaf (IC_50_ = 117.20 ± 4.84 µg/mL) extracts exhibited high elastase inhibitory activity at similar levels. *R. japonica* stem extract (IC_50_ = 253.03 ± 3.05 µg/mL) had moderate elastase inhibitory activity, while its root extract (9.31%) exhibited low elastase inhibition. *R. japonica* leaf (IC_50_ = 171.00 ± 6.76 µg/mL) and root (IC_50_ = 160.00 ± 6.81 µg/mL) extracts had parallel and remarkable collagenase inhibitory activity, while its flower (IC_50_ = 231.90 ± 4.12 µg/mL) and stem (IC_50_ = 236.80 ± 7.34 µg/mL) extracts were found to be moderately and similarly effective (Table 2).

### 2.2. Results of Phytochemical Analyses

Sample data were ion-extracted using molecular feature extraction (MFE) and sorted via abundance. Most abundant peaks were investigated at the first step. Both precursor (M-H) and product ion values were checked for the identification of compounds. Databases, literature, and certified references matched identities, and a list of identified compounds in our samples is given in detail in Table 3 [5,21]. Procyanidine dimer, catechin, piceid (resveratrol glucoside), torachrysone glucoside, and torachryson isomers were found as major components in the *R. japonica* extracts. One representative RJK sample chromatogram is given below (Figure 1). All chromatograms and mass spectra of compounds can be found in Appendix A.

According to our findings, it was observed that different parts of *R. japonica* extracts contained similar compounds and the same major components. The root extracts have also been determined to possess more affluent and more intense content than other plant parts (flowers, stems, and leaves). The comparative chromatogram of the extracts from *R. japonica* roots, stems, flowers, and leaves is given below (Figure 2).

The phytochemical analysis indicated small amounts of resveratrol to be present in *R. japonica* extracts. Even though the highest amount of resveratrol was found in the *R. japonica* root extract, it was not detected in the stem extract (Figure 3).

### 2.3. Ligand Binding Energy Analysis Across Proteins

Based on the experimental results we obtained, selected ligands (galloylglucose, emodin, emodin-6-glucoside, and piceid) identified as the major components in the extracts were analyzed through computational approaches targeting AChE, BChE, collagenase, and elastase. The aim was to estimate the free energy released upon ligand binding, providing insights into their likely stability within the active sites. Since multiple poses exist for each complex, there are multiple binding energies as well. To illustrate the complete findings, box and violin plots were used to present the docking scores, as shown in Figure 4. For AChE, emodin-6-glucoside exhibited the lowest binding energies, ranging approximately between −10.5 and −9.0 kcal/mol, suggesting a high binding affinity and stability in the active site. This contrasts with emodin, whose binding energies are distributed in the range of approximately −9 to −6 kcal/mol, with higher variability observed for emodin.

Similarly, for BChE, emodin-6-glucoside again indicated a trend toward more stable binding energies, with values extending as low as −10 kcal/mol. However, galloylglucose and emodin also demonstrated competitive binding energies, with median values hovering around −6 to −8 kcal/mol. The wider distribution for emodin compared to galloylglucose indicates a broader range of poses within the active site, reflecting higher flexibility.

In the case of collagenase, both piceid and emodin-6-glucoside were revealed to have comparable binding energy distributions. Most values lie between −7.5 and −6.0 kcal/mol, with piceid showing slightly lower minima. The narrow spread and dense clustering of energies indicate consistent and stable binding for both ligands, suggesting their high affinity and structural compatibility with the active site of collagenase.

For elastase, the binding energies of emodin-6-glucoside were observed to cluster tightly around −6 to −7 kcal/mol, with lower variability compared to emodin. Emodin exhibited a broader range of energies, extending from approximately −8 to −4 kcal/mol, which indicates greater flexibility or diversity of binding poses. The tighter distribution of emodin-6-glucoside suggests a more stable and predictable interaction with elastase’s active site.

Among all ligands and proteins examined, emodin-6-glucoside consistently demonstrated lower binding energies, particularly with AChE and BChE, indicating its strong potential as a stabilizing ligand. In contrast, emodin showed greater variability across all proteins, reflecting its structural flexibility and diverse binding poses. Notably, piceid was only examined with collagenase, where it displayed competitive stability.

Overall, emodin-6-glucoside emerged as the most promising ligand due to its consistently lower binding energies and tighter distributions, particularly with AChE and BChE. The observed variability in other ligands, such as emodin, indicates their ability to adapt to multiple conformational states, which may be advantageous in some contexts but could reduce binding predictability.

The 2D interaction diagrams as presented in Figure 5 illustrate the binding of the best-performing ligands, e.g., galloylglucose (AChE and BChE), piceid (collagenase), and emodin (elastase) to their respective active sites. The diagrams depict hydrogen bonds (red lines) and polar interactions between the ligands and amino acid residues, highlighting the key residues stabilizing these ligands within the active sites. Galloylglucose at AChE (a): Galloylglucose forms multiple stabilizing interactions within the active site of AChE. Key hydrogen bond interactions are observed with residues ARG 296, ASP 74, THR 83, and TYR 124. Additional interactions include hydrophobic and polar contacts with residues such as PHE 295, PHE 297, TRP 286, TYR 337, and PHE 338. The strong involvement of ARG 296 and ASP 74 suggests that these residues play a critical role in anchoring the ligand, stabilizing it via electrostatic and hydrogen bond interactions. The dense network of interactions indicates a strong and stable binding pose, contributing to the observed low binding energy. Galloylglucose at BChE (b): In BChE, galloylglucose demonstrates strong hydrogen bonding with key residues, including ASN 83, THR 120, GLN 119, and ASP 70. Notably, polar interactions also occur with residues SER 198, HIP 438, and TRP 82, while additional stabilizing contacts involve PHE 329 and TYR 332. The presence of multiple hydrogen bonds, particularly with ASP 70 and THR 120, reinforces the ligand’s stabilization within BChE’s active site. Compared to AChE, the hydrogen bond network here appears more evenly distributed across the active site residues, reflecting a well-anchored and flexible binding mode. Piceid at collagenase (c): Piceid interacts strongly with the active site of collagenase through several critical hydrogen bonds, including GLU 219, ASN 180, HIS 218, and THR 241. A notable interaction also occurs with a water molecule (H_2_O), suggesting its role as a mediator in ligand stabilization. Other residues, such as VAL 215, PRO 238, and TYR 240, contribute to additional hydrophobic and polar interactions. The hydrogen bonds with GLU 219 and HIS 218 are particularly significant, as these residues may play a key role in catalytic or stabilizing functions within the active site. The involvement of a water bridge highlights the importance of solvent effects in ligand binding, further stabilizing the ligand within the collagenase active site. Emodin at elastase (d): Emodin exhibits fewer hydrogen bonds compared to the other ligands but maintains strong polar interactions with ASN 147 and ASP 194. Additional stabilizing contacts occur with residues SER 195, SER 217, GLY 216, and CYS 220. Notably, hydrophobic interactions with PHE 215 and VAL 213 contribute to the ligand’s overall stability within the active site. The reduced number of hydrogen bonds compared to galloylglucose and piceid suggests that hydrophobic interactions play a more significant role in stabilizing emodin within elastase. Despite this data, the ligand demonstrates a well-defined binding pose with key residues such as ASN 147 and ASP 194 forming critical contacts.

### 2.4. Drug-Likeness Parameters and ADME Profile Analysis of Galloylglucose, Emodin, Emodin-6-Glucoside, and Piceid

The optimal physicochemical parameters for oral bioavailability can be characterized within a defined multidimensional space (see Figure 6 and Figure 7). The lipophilicity, quantified by XLOGP3, should fall within the range of −0.7 to +5.0, representing a balanced partition coefficient that facilitates both membrane permeability and aqueous solubility. The molecular size, expressed as molecular weight (MV), is constrained between 150 g/mol and 500 g/mol, which aligns with traditional guidelines for drug-like molecules [22]. The molecular polarity, measured by topological polar surface area (TPSA), should be maintained between 20 Å^2^ and 130 Å^2^, ensuring appropriate membrane permeation while maintaining sufficient aqueous solubility. The aqueous insolubility parameter, expressed as Log S (ESOL), should range from −6 to 0, indicating acceptable solubility characteristics in physiological environments. Molecular saturation, represented by the fraction of sp^3^ carbons (Fraction Csp3), should be maintained between 0.25 and 1, reflecting an appropriate balance between aromatic and saturated carbon centers. Finally, molecular flexibility, quantified by the number of rotatable bonds, should be limited to between 0 and 9, optimizing the conformational entropy contribution to binding energy, while maintaining appropriate oral absorption characteristics.

The molecular weights range from 240.21 g/mol (emodin) to 388.32 g/mol (emodin-6-glucoside). Emodin, being the aglycone, displays the simplest structure with 18 heavy atoms, while its glycosylated form and other compounds contain 21–28 heavy atoms. The Topological Polar Surface Area (TPSA) varies significantly, with emodin having the lowest value (74.60 Å^2^) and galloylglucose showing the highest (156.91 Å^2^). Galloylglucose demonstrates exceptional water solubility (Log S ESOL: −0.77), categorized as “very soluble” with 51.4 mg/mL. Emodin shows the lowest water solubility (Log S ESOL: −3.81) among the compounds, though still classified as “soluble” with 0.037 mg/mL. The glycosylation pattern notably improves water solubility, as seen in emodin-6-glucoside and Piceid. The consensus Log Po/w values reveal that emodin is most lipophilic (1.98), Piceid is moderate (0.39), while Emodin-6-glucoside (−0.18) and galloylglucose (−1.43) have hydrophilic character. Boiled-EGG method also shows that emodin has a high gastrointestinal (GI) absorption capacity, while galloylglucose, emodin-6-glucoside, and piceid have a low absorption, while this pattern suggests that glycosylation generally reduces GI absorption. Only emodin shows blood–brain barrier (BBB) permeation capability, while all glycosylated compounds are non-permeable.

While traditional drug-likeness rules such as Lipinski’s Rule of Five (MW < 500; MLOGP < 4.15; N or O < 10; NH or OH < 5) have been widely used to evaluate oral bioavailability for over two decades, recent research has identified numerous exceptions to these rules, particularly for natural products and their derivatives [23]. Emodin fully complies with all Lipinski criteria, suggesting favorable physicochemical properties for oral administration. Each of piceid and galloylglucose violated one rule (NH or OH > 5), suggesting reduced permeability and potential challenges in passive absorption. Emodin-6-glucoside is also fully compliant, implying a favorable oral bioavailability profile. However, these classical rules should be interpreted with caution, as modern drug discovery has validated many successful drugs that violate these parameters [24].

Expanded evaluation using alternative filters (Ghose, Veber, Egan, and Muegge) provided additional insights into the compounds’ potential therapeutic applications. With the exception of emodin, all other compounds demonstrated limited bioavailability due to multiple rule violations, reinforcing potential challenges in their oral absorption. Glycosylated compounds (piceid, galloylglucose, and emodin-6-glucoside) exhibited violations primarily related to topological polar surface area (TPSA) thresholds, which may indicate limited membrane permeability. These findings suggest that alternative delivery strategies, such as topical applications or specialized drug delivery systems, might be more appropriate for the glycosylated derivatives, while emodin appears more suitable for conventional oral administration. These ADME profile differences directly influence potential therapeutic applications, with glycosylated compounds potentially more suitable for local applications (such as skin treatments for elastase/collagenase inhibition) and emodin potentially more appropriate for systemic conditions requiring BBB penetration.

### 2.5. Cytochrome P450 Inhibition and Potential Drug Interactions

The interaction of the selected compounds with cytochrome P450 (CYP) enzymes was evaluated to assess potential metabolic liabilities. Emodin exhibited inhibitory potential towards CYP1A2 and CYP3A4, suggesting a risk for drug–drug interactions that may influence the metabolism of co-administered substrates of these enzymes. In contrast, piceid, galloylglucose, and emodin-6-glucoside did not demonstrate significant CYP inhibition, indicating a lower likelihood of CYP-mediated interactions.

Pan-assay interference compounds (PAINS), described as the computational analyses implemented by Baell and Holloway [25], alerts indicate that certain structural features within the molecules may lead to non-specific interactions, resulting in false-positive results in biological assays rather than genuine target-specific activity. These alerts emphasize the necessity for further experimental validation to distinguish genuine pharmacological and toxicological effects from potential assay artifacts. Screening through PAINS led to the identification of structural features associated with non-specific bioactivity. Emodin and emodin-6-glucoside triggered a quinone_A alert, indicating the presence of a quinone moiety, which may lead to redox cycling and non-specific reactivity. Galloylglucose was flagged for a catechol_A alert, suggesting potential interference due to its polyphenolic structure. Piceid exhibited a stilbene alert, which may be relevant to its biological activity and potential interactions.

### 2.6. Synthetic Accessibility Assessment

The feasibility of chemical synthesis was evaluated using synthetic accessibility (SA) scores, where lower values indicate easier synthesis. The scores ranged from 2.38 (emodin, most accessible) to 4.76 (emodin-6-glucoside, most challenging). The higher SA scores of glycosylated derivatives suggest increased synthetic complexity, likely due to the challenges associated with regioselective glycosylation and purification. These findings provide a comprehensive overview of the pharmacokinetic, drug-likeness, and medicinal chemistry properties of the selected compounds, highlighting both their therapeutic and toxicological potential and structural liabilities.

### 2.7. Statistical Analyses

The proportion of enzyme inhibitory effects was statistically analyzed using one-way ANOVA, followed by Dunnett’s multiple comparison test to compare the positive control with the test groups (GraphPad Prism 6.01). Values of *p* ≤ 0.05 were deemed statistically significant.

## 3. Discussion

Medicinal plants can play a role in treating diseases with their bioactive components and different action mechanisms [26]. Finding bioactive ingredients from medicinal plant sources is a critical way to develop new drug candidates. In the case of disease, the expression of some enzymes may be abnormally increased or decreased. For this reason, enzymes have become the main target as a research strategy for many important diseases, such as AD, diabetes, hyperpigmentation, and cancer [27]. The discovery and evaluation of enzyme inhibitors have attracted intense interest from researchers in many fields, such as endocrinology, pharmacology, and toxicology [28].

*R. japonica*, whose enzyme inhibitory activities were evaluated in this study, contains many noteworthy phytochemicals, principally resveratrol, and is frequently investigated for its biological activities. The roots of *R. japonica* are widely used in China and Japan for treating diseases such as inflammation, infection, jaundice, skin burns, and hyperlipemia [29]. In Türkiye, *R. japonica* was diagnosed in Samsun Terme/Bazlamaç region in 2019, and with the addition of this genus, the number of genera of the Polygonaceae family in the flora of Türkiye increased to 11 [9]. In a study, the ethyl acetate extract of *R. japonica* roots showed a notable inhibitory activity against AChE (IC_50_ = 27.04 ± 1.18 µg/mL) and BChE (IC_50_ = 11.29 ± 1.54 µg/mL). *R. japonica* root ethanol extract moderately inhibited AChE (26.71%) and BChE (16.00%) [30]. In the current study, *R. japonica* stem (AChE: IC_50_ = 25.06 ± 0.52 µg/mL; BChE: IC_50_ = 159.78 ± 3.50 µg/mL), leaf (AChE: IC_50_ = 53.31 ± 3.38 µg/mL; BChE: IC_50_ = 202.50 ± 1.41 µg/mL), flower (AChE: IC_50_ = 64.44 ± 4.69 µg/mL; BChE: 41.13%), and root (AChE: IC_50_ = 120.10 ± 0.14 µg/mL 49.93%; BChE: 20.15%) extracts inhibited AChE and BChE in varying levels. Compared to our study, the anti-BChE activity of *R. japonica* root ethanol extract was similar, while the anti-AChE activity was lower than our results. The antioxidant activity, anticholinesterase effects, and phenolic substances were analyzed via HPLC in aqueous-methanol extracts obtained from different parts of the *R. japonica* plant indigenous to Serbia. Extracts from leaves and rhizomes were abundant in chlorogenic and rosmarinic acid. However, resveratrol was identified only in the rhizomes and stem extracts. The evaluation of AChE inhibition among all extracts revealed that the rhizome extract had the most significant effect (79.05% at 1260 µg/mL), which contradicts our findings [20]. The variation in enzyme inhibition data may be attributed to the distinct phytochemical profiles and the differing concentrations employed in the method. Despite the observation of dose-dependent inhibition in studies examining the AChE inhibitory effects of extracts from another *Polygonum* species (*Polygonum hydropiper* L.) and essential oils derived from its leaves and flowers, phylogenetic analyses have established that this species belongs to a different genus [*Persicaria hydropiper* (L.) Delarbre] [31,32]. In another study, *R. japonica* root extract had no inhibitory activity on AChE. In the same study, *R. japonica* root inhibited TYR by 20.24% to 94.21% (104.2 µg/mL–833.3 µg/mL) [33]. In fact, many anti-TYR activity studies can be found on *R. japonica,* along with the isolation of its compounds. Four anthraquinones (physcion, emodin, citreorosein, and anthraglycoside B) isolated from *R. japonica* roots presented moderate or strong TYR inhibition, whereas two stilbenes, e.g., resveratrol and piceid, showed no anti-TYR activity [34]. In a previously reported study, piceid isolated from *R. japonica* displayed a concentration-dependent anti-TYR activity (80%, at a concentration of 50 µg/mL) and suppressed melanin synthesis [35]. TYR activity (22.6%) and melanin synthesis in B16-F10 cells were reduced by *R. japonica* [36]. Our TYR inhibition assay pointed out that *R. japonica* root (19.46%) and stem (26.03%) had a moderate inhibitory effect, but no activity was observed with its leaves and flowers. As aforementioned, our results are consistent with those of relevant studies reported up to date. A study in France designed ultrasonication-assisted ethanol extracts from the aerial parts and roots of *R. japonica*, examining the dermocosmetic effects of these extracts, including their inhibitory effects on hyaluronidase, elastase, collagenase, and TYR. Moreover, the enzyme inhibition of some naturally derived compounds in the extracts has been evaluated. Concurrently with our findings, the extracts exhibited elevated anticollagenase activity and diminished anti-TYR effects. A moderate inhibition of hyaluronidase was discovered, although a modest level of elastase inhibition was detected. Upon examination of the phytochemical profiles of the extracts, it was noted that the root extract, characterized by a higher concentration of stilbene and quinone derivatives, had more activity in enzyme inhibition assays. Upon examining the enzyme inhibition results of natural phytochemicals in separation, it was established that these compounds may exert a synergistic impact within the extract [37]. The anti-elastase activity of the *R. japonica* aqueous extract and the fraction obtained by polyamide column chromatography were found to be 53.56% and 61.27%, respectively. Polydatin (piceid), the extract’s main component, showed 82.53% elastase inhibition [38]. In this study, *R. japonica* root extract (IC_50_ = 160.00 ± 6.81 µg/mL) possessed higher anti-collagenase activity than other parts of the plant but weak anti-elastase activity. In our study, *R. japonica* flower (IC_50_ = 111.40 ± 1.45 µg/mL) and leaf extracts (IC_50_ = 117.20 ± 4.84 µg/mL) had the highest anti-elastase activity. However, *R. japonica* plant parts were found to have a lower effect compared to the synthetic positive controls used for anti-collagenase (1,10-phenanthroline; IC_50_ = 27.95 ± 0.37 µg/mL) and anti-elastase [*N*-(methoxysuccinyl)-Ala-Ala-Pro-Val-chloromethyl ketone; IC_50_ = 2.65 ± 0.36 µg/mL] activity studies.

It should also be noted that in enzyme inhibition studies with plant extracts, it is necessary to determine the main compounds responsible for the activity by activity-guided isolation study. In a phytochemical study on *R. japonica*, emodin, phalacinol, emodin-8-*O*-beta-D-glucopyranoside, 20-methoxy-6-acetyl-7-methylyuglone, resveratrol, and piceid were isolated. In addition, *cis*-resveratrol and *cis*-piceid were obtained by phytochemical isomerization in the same study [39]. In a phytochemical analysis, 53 metabolites were elucidated in *R. japonica,* most of which were stilbenes (resveratrol and glucosides), anthraquinones (emodin and its derivates), and phenolic compounds (luteolin, epigallate catechin gallate, apigenin-6-glucoside, kaempferide, formononetin 7-*O*-glucoside, bergapten, 6-malonylgenistin and procyanidin-B-1,3*-O*-gallate) [21]. In another study, *R. japonica* was reported to contain mainly anthraquinones, stilbenes, and torachryson together with their derivatives covering resveratroloside, polydatin, emodin-8-*O*-glucoside, resveratrol, toracryson-8-*O*-glucoside, emodin-1-*O*-glucoside, toracryson-8-*O*-(6′-acetyl)glucoside, physcion-8-*O*-glucoside, physcion-8-*O*-(6′-acetyl)glucoside, and emodin [40]. In our study with LC-MS QTOF, the main components of *R. japonica* were determined as piceid, thoracrysone isomer, thoracrysone glucoside, procyanidin dimer, and catechin. It was found to have similar contents to the phytochemical studies previously conducted on *R. japonica*. It has been determined in previous studies that the roots of *R. japonica* contained more resveratrol and resveratrol glycosides as compared to its leaves and stems [41]. In a phytochemical analysis by HPLC, the resveratrol content of *R. Japonica*’s perennial root, leaf, stem, and annual root was determined as 1024.96, 764.74, 123.57, and 26.88 μg/g, respectively [42]. Like previous studies, the highest resveratrol content was revealed in the roots in our study, but unlike other studies, resveratrol was not detected in the stem parts. In addition, resveratrol, specified as the main component of *R. japonica* in the literature, was detected at low levels in the studied plant parts of the plant sample collected from Türkiye. Namely, our study, which has been the first phytochemical and bioactivity investigation of *R. japonica* of Turkish origin, led to the identification of piceid (a resveratrol glycoside) as the major component in all extracts.

Docking results revealed that galloylglucose displayed the strongest binding at AChE and BChE, stabilized by key hydrogen bonds with residues such as ARG 296 and ASP 70. Piceid formed significant interactions with collagenase, including water-mediated contacts, while emodin interacted with elastase through polar and hydrophobic contacts, notably with ASN 147 and ASP 194. These interactions align with binding energies, highlighting the importance of hydrogen bonds and hydrophobic interactions for ligand stability and protein inhibition.

The computational ADME, pharmacokinetics, and toxicokinetic analyses exposed distinctive characteristics of these compounds. Emodin was determined to have the most favorable drug-like properties but poses potential interaction risks. The glycosylated derivatives displayed improved water solubility versus reduced absorption. These findings suggested different potential applications: Emodin might be suitable for oral administration, while its glycosylated forms might be better suited for topical or targeted delivery systems. In toxicokinetic analyses, several alerts were revealed. For instance:Quinone_A Alert (emodin and emodin-6-glucoside): Quinone groups can undergo redox cycling, leading to the generation of reactive oxygen species (ROS), which may cause cytotoxicity independent of a specific biological target.Catechol_A Alert (galloylglucose): Catechol moieties have a known tendency to chelate metal ions and inhibit various enzymes non-selectively, potentially leading to misleading bioactivity results.Stilbene Alert (piceid): Stilbene derivatives can exhibit aggregation behavior or auto-fluorescence, which may interfere with fluorescence-based assays, distorting the interpretation of biological activity.

Given these considerations, additional studies are required to validate the biological relevance of these interactions and ensure that observed pharmacological and toxicological effects in future studies are not simply artifacts arising from the assay conditions. Without such verification, the therapeutic potential of these compounds may be overestimated, or their toxicity profile may be inaccurately characterized. Emodin seems to be exhibiting a potential inhibitory effect on cytochrome P450 1A2 (CYP1A2), suggesting possible pharmacokinetic interactions with drugs metabolized by this enzyme. While given the absence of significant CYP inhibition for galloylglucose, emodin-6-glucoside, and piceid, their impact on drug metabolism is expected to be minimal. However, glycosylation may influence prodrug activation or bioavailability [43], which warrants further investigation. 

## 4. Material and Methods

### 4.1. Plant Material and Extraction Method

*R. japonica* ([Fig pharmaceuticals-18-00408-p001]) was collected from Samsun province (Türkiye) in 2021 (A6 Samsun; Terme, Bazlamaç district, field margins, 250 m, 26 August 2021, O. Tugay 19.035 & A. Özbek) and identified by Prof. Dr. Osman Tugay. The plant material was divided into four parts, i.e., stem, leaf, flower, and root. After the addition of ethanol to each plant part, they were left to macerate occasionally shaking by hand at room temperature for 5 days. Then, they were filtered through filter paper and concentrated at 40 °C by a rotary evaporator (Buchi, Flawil, Switzerland). The extracts were kept in the refrigerator at 4 °C until the bioassays and phytochemical analyses were carried out.

### 4.2. Enzyme Inhibition Assays

#### 4.2.1. AChE/BChE Inhibition Assay

Ellman’s approach was modified to assess the inhibitory activity of the extracts against the AChE/BChE enzymes [44]. AChE (Type-VI-S, EC 3.1.1.7, Sigma, Burlington, MA, USA) from the electric eel (*Electrophorus electricus*) and BChE (EC 3.1.1.8, Sigma) from equine serum were used as the enzyme sources. Acetylthiocholine iodide/butyrylthiocholine chloride was used as the reaction substrates. The cholinesterase activity was assessed using 5,5-dithio-bis (2-nitrobenzoic) acid. A total of 140 µL of 0.1 mM sodium phosphate buffer (pH 8.0) was first added to a 96-well microplate. Next, 20 µL of the extracts/control were added, which was followed by the addition of 0.2 M enzyme solution. It was incubated for 10 min at room temperature. To begin the reaction, the 96-well microplate was filled with 10 µL of 0.2 M acetylthiocholine iodide/butyrylthiocholine chloride as the substrates. Thiocholine is produced when AChE or BChE hydrolyzed thiol esters are employed as substrates. Thiocholine reacts with DTNB to produce the yellow substance 2-nitro-5-thiobenzoate (TNB), which is the end product of the reaction. Utilizing an ELISA microplate reader (Molecular Devices, Spectramax i3x microplate reader with Softmax^®^ Pro Software for Windows 10 (San Jose, CA 95134, USA) at a wavelength of 412 nm. The reference drug in both assays was galantamine hydrobromide (Sigma, Burlington, MA, USA).

#### 4.2.2. Elastase Inhibition Assay

The spectrophotometric technique was developed by Kraunsoe et al. and refined by Lee et al. [45,46] was used herein to investigate the inhibition of elastase. *N*-Suc-(Ala)3-*p*-nitroanilide was the substrate, and porcine pancreatic elastase (Type IV, Sigma, EC 3.4.21.36) was the enzyme source. The procedure is used based on the measurement of nitroaniline emitted from the substrate at a wavelength of 410 nm with an ELISA microplate reader. In a 96-well microplate, 1.015 mM substrate was produced in Tris-HCl buffer (0.1 M, pH 8.0) and combined with the extracts dissolved in DMSO. The enzyme solution was introduced to the microplate after a five-minute pre-incubation period at 25 °C with 0.5 unit/mL (15 µL) of the enzyme. The reference substance was *N*-(Methoxysuccinyl)-Ala-Ala-Pro-Val-chloromethyl ketone.

#### 4.2.3. Collagenase Inhibition Assay

Collagenase inhibition was measured using the modified spectrophotometric method developed by Wart and Steinbrink [47,48]. *Clostridium histolyticum* (Sigma, EC 3.4.24.3) was the enzyme source, while *N*-(3-[2-furyl]acryloyl)-Leu-Gly-Pro-Ala (FALGPA) was used as the substrate. A total of 0.8 units/mL of the enzyme was dissolved in 50 mM tricine buffer (pH 7.5), and the substrate FALGPA was dissolved in the same buffer. Buffer, DMSO/extract, and the enzyme were mixed in a 96-well microplate well. After 15 min of pre-incubation, 50 µL of the substrate was added. The absorbance was read at 340 nm for 20 min at 2 min intervals in an ELISA microplate reader 1,10-phenanthroline was used as the reference.

#### 4.2.4. TYR Inhibition Assay

L-DOPA was used as a substrate for the spectrophotometric approach to measure TYR inhibition [46]. The strategy is based on measuring the absorbance of dopachrome, which is created when the enzyme and substrate react at a wavelength of 492 nm. The 96-well microplate was filled with either a sample solution or DMSO. Then, 5 mM L-DOPA and 67 mM phosphate buffer (pH 6.8) were added. A total of 30 µL of mushroom TYR (Sigma, EC 1.14.18.1) enzyme solution was added after 10 min of incubation at 37 °C. The absorbance at 492 nm was measured using an ELISA microplate reader. The reference was *alpha*-kojic acid.

#### 4.2.5. Calculation of Data for Enzyme Assays

Each sample was run in four parallel experiments in the AChE/BChE, elastase, collagenase, and TYR inhibition assays, and the findings are shown as the mean and standard deviation (S.D.) of inhibitions% from four experiments. To calculate their IC_50_ values, GraphPad Prism 8.0 was used. The sample percentages of enzyme inhibition were determined using the formula below.Inhibition% = 100 − [(A_1_/A_2_) 100].

The sample solution absorbance is A_1_. The average absorbance of the adverse control solutions is A_2_.

### 4.3. Conditions of LC-MS QTOF Analysis

The samples were dissolved in 1 mL methanol and analyzed via liquid chromatography quadrupole time of flight mass spectrometer (LC-MS Q-TOF). Agilent 1290 Series HPLC (Santa Clara, CA, USA) with a binary pump, high-performance HiP autosampler, and column thermostat coupled with an Agilent G6530B. A quadrupole time-of-flight LC-MS system was used for the analyses. Dual electrospray ionization (ESI) source in negative mode were employed. The instrument was operated in a 2 GHz extended dynamic range mode. All ion modes with collision energies 0 eV, 10 eV, and 30 eV were used during the run. The analytical column used for separation was Agilent Poroshell SB C-18 (3.0 mm × 100 mm × 2.7 µm). Details for the instrumental conditions are shown in Table 4. The data was processed by Agilent MassHunter software B 07.00. Compounds were identified using analytical reference standards and a library search using the Metlin Metabolite database, Massbank.eu, and PubChem.

### 4.4. Optimization Pipeline for Protein Crystal Structures and Ligands

The crystal structures of AChE, BChE, collagenase, and elastase were obtained from the Protein Data Bank (PDB) using the codes 4EY7 [49], 5DYW [50], 1CGL [51], and 1BRU, respectively. These structures require preparation prior to use, including the addition of hydrogen atoms (which are absent in the original structures) and other processes that will be described subsequently. Schrödinger Maestro, a software package equipped with various tools for working with protein structures, was used to carry out this process. The Protein Preparation Wizard [52] in Maestro was employed to assign bond orders, create disulfide bonds, and remove water molecules located more than 5 Å from HET groups. The system was adjusted to biological pH using the integrated PROPKA tool [53]. Finally, the entire system was minimized using the OPLS3 force field [54] to resolve atomic clashes. The ligand structures in structure data file (SDF) format were obtained from PubChem and underwent energy minimization under biological pH conditions using Epik implemented in LigPrep [55].

### 4.5. Ligand Binding Energy Analysis Using Induced Fit Docking

Ligand binding energies within the protein were calculated using the induced fit docking (IFD) method [56,57,58]. This approach allows both the ligand and the active site to adapt their positions and conformations during simulations, enabling the generation of multiple plausible ligand poses within the binding domain. Each pose was assigned a corresponding binding energy. The main difference between IFD and standard docking is that, in each simulation round, IFD refines the side chains of the active site amino acids surrounding the ligand. In contrast, the standard docking method keeps the target atoms rigid throughout the simulation. If the crystal structure contains any unusual conformations due to the crystallographic process, the ligand may not fit properly into the binding site, reducing the accuracy of the results (Appendix A).

### 4.6. Computational ADME, Pharmacokinetic and Toxicokinetic Predictions

All physicochemical properties, pharmacokinetics, drug-likeness, and medicinal chemistry parameters were calculated using the SwissADME web tool, developed and maintained by the Molecular Modeling Group of the Swiss Institute of Bioinformatics [59]. This platform implements various validated predictive models and algorithms to generate a comprehensive profile of the selected compounds.

Through SwissADME, the topological polar surface area (TPSA) was computed using the method described by Ertl et al. [60]. Lipophilicity was assessed using multiple predictive models for log P_o/w implemented within the platform, including the physics-based iLOGP designed to reduce overfitting bias, and other data-driven approaches trained on extensive experimental datasets. A consensus approach was adopted, where agreement among multiple predictors increases confidence in the predicted log P_o/w value.

For toxicokinetic predictions, we utilized the pan-assay interference compounds (PAINS) alerts integrated within SwissADME, which identify structural features that may lead to non-specific interactions or false-positive results in biological assays. Additionally, the platform’s implementation of Brenk filters helped identify potentially toxic or reactive functional groups. These toxicokinetic predictions highlight potential safety concerns and provide crucial information for future development strategies.

Pharmacokinetic parameters, such as P-glycoprotein (P-gp) substrate status and CYP450 inhibition potential, were predicted using Support Vector Machine (SVM) models embedded in SwissADME. These binary classification models estimate the likelihood of a molecule being a P-gp substrate or a CYP450 inhibitor by comparing its molecular descriptors with those of known substrates and inhibitors. The predictive performance of these models has been validated through 10-fold cross-validation and external test sets, making them suitable for early-stage drug discovery.

The skin permeability coefficient (K_p_) was predicted using the linear model developed by Potts and Guy [61] as implemented in SwissADME. In this model, log K_p_ values indicate the passive permeability of a molecule through the mammalian epidermis, with more negative values corresponding to lower permeability.

Additionally, passive gastrointestinal (GI) absorption and BBB permeability were assessed using the Brain Or IntestinaL EstimateD permeation (BOILED-Egg) model [62] within the SwissADME platform. This model evaluates a molecule’s likelihood of intestinal absorption and BBB penetration based on its lipophilicity (WLOGP) and apparent polarity (TPSA). The platform’s visualization tool places molecules predicted to have high GI absorption within the white region of the BOILED-Egg plot, while those likely to cross the BBB fall within the yellow region (yolk). The model also accounts for P-gp efflux, with color-coded markers indicating active efflux by P-gp.

A limitation of these computational predictions is the absence of in vivo validation, which would be necessary to confirm the predicted pharmacokinetic and toxicokinetic profiles. Future studies should include experimental validation of these computational findings to establish more definitive ADME profiles for these compounds.

## 5. Conclusions

*R. japonica*, which is commonly known as *P. cuspidatum*, is a medicinal plant used in traditional Chinese medicine. Likewise, *P. ciliinerve* and *P. multiflorum*, which contain resveratrol, were also found to be transferred to the genus *Reynoutria* by the latest phylogenetic verifications. *R. japonica* extracts markedly inhibited cholinesterases, TYR, elastase, and collagenase. Although we identified chromatographic characteristics of the extracts, the main component(s), which might be responsible for the mentioned enzyme inhibitory effects, can only be determined by the activity-guided isolation technique. In this study, we have revealed that *R. japonica* grown in Türkiye can be a source of potential resveratrol derivates and other components. Our findings also emphasize that the active inhibitory extracts towards AChE/BChE can be considered to proceed for further research on AD. Further, in vivo studies may be conducted to explore the neuroprotective effects of the extracts and/or the compounds responsible for this effect.

## Figures and Tables

**Figure 1 pharmaceuticals-18-00408-f001:**
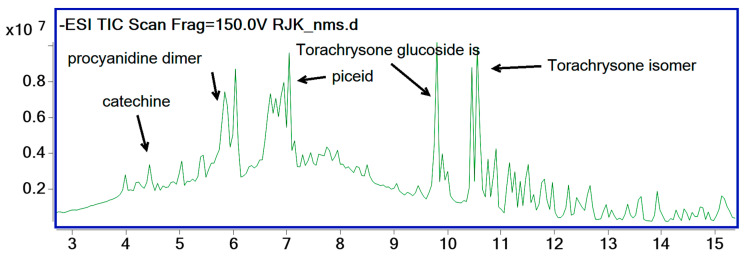
A representative TIC of sample chromatogram showing major compounds.

**Figure 2 pharmaceuticals-18-00408-f002:**
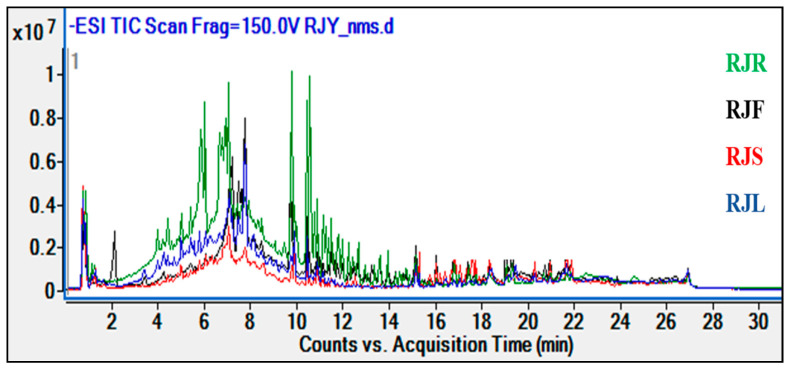
Comparative chromatogram of *R. japonica* extracts.

**Figure 3 pharmaceuticals-18-00408-f003:**
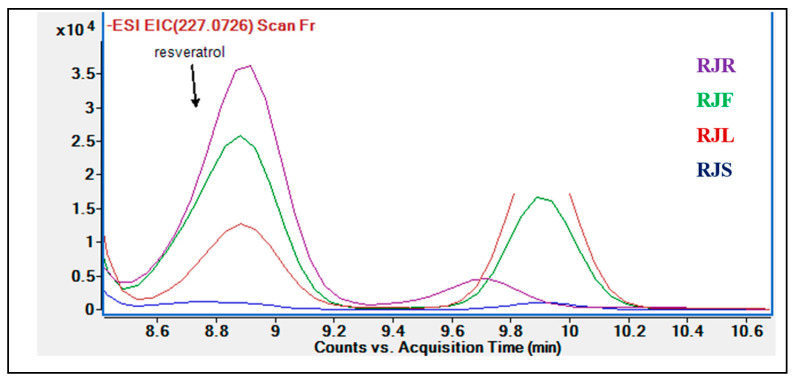
Comparative chromatogram of *R. japonica* extracts based on their resveratrol contents.

**Figure 4 pharmaceuticals-18-00408-f004:**
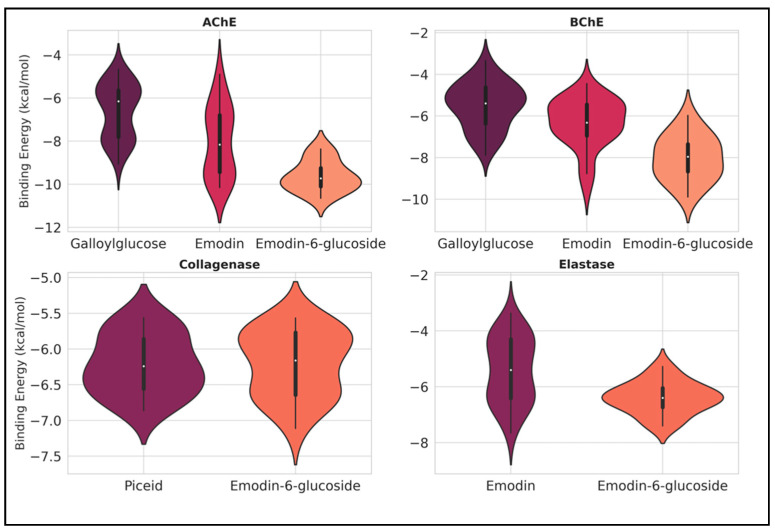
Violin plots showing the distribution of binding energies (kcal/mol) for different ligands against AChE, BChE, collagenase, and elastase proteins. Each plot illustrates the range, density, and variability of binding energies, with lower values indicating higher binding affinity and stability.

**Figure 5 pharmaceuticals-18-00408-f005:**
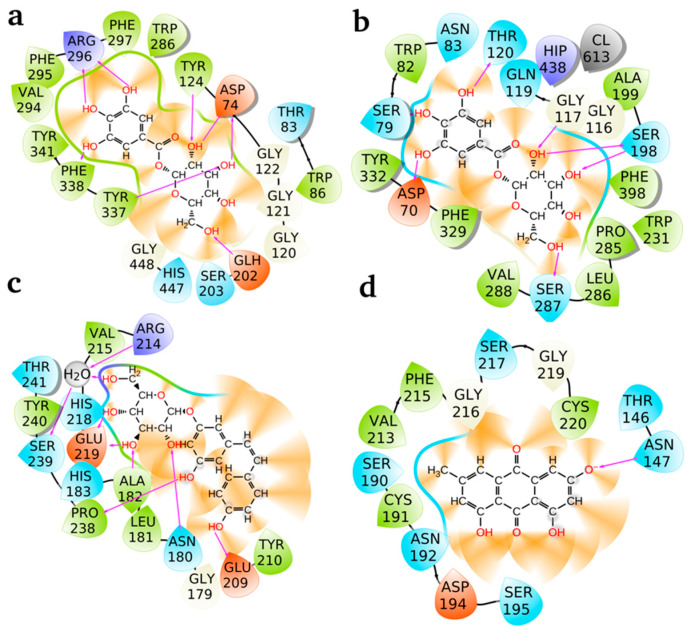
Two-dimensional interaction diagrams of the best ligands interacting with active site residues of target proteins: (**a**) Galloylglucose at AChE; (**b**) Galloylglucose at BChE; (**c**) Piceid at collagenase; and (**d**) Emodin at elastase. Pink lines represent hydrogen bonds, while other polar and hydrophobic interactions with amino acid residues are shown. Key residues stabilizing the ligands are highlighted, demonstrating the interactions contributing to ligand binding affinity and stability within the active sites.

**Figure 6 pharmaceuticals-18-00408-f006:**
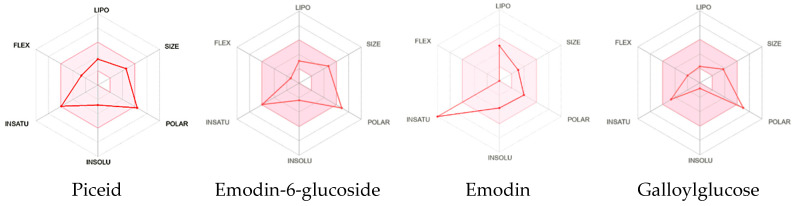
The colored zone is a suitable physicochemical space for oral bioavailability. LIPO (lipophilicity): −0.7 < XLOGP3 < 5; SIZE: 150 g/mol < MV < 500 g/mol; POLAR (polarity): 20 A < TPSA < 130 A; INSOLU (insolubility): −6 < Log S (ESOL) < 0; INSATU (insaturation): 0.25 < Fraction Csp3 < 1; FLEX (flexibility): 0 < number of rotatable bonds < 9.

**Figure 7 pharmaceuticals-18-00408-f007:**
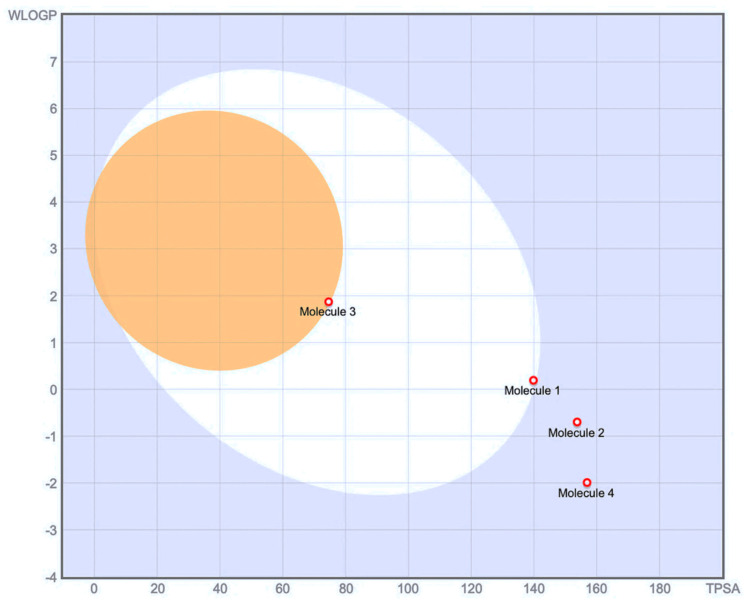
Molecules located in BOILED-Egg’s yolk (orange area) are molecules predicted to passively permeate through the blood-brain barrier. Molecules located in the white area are molecules predicted to be passively absorbed by the gastrointestinal tract. The light blue background represents the area where molecules are neither absorbed by the GI tract nor cross the blood-brain barrier. Red dots are molecules predicted not to be effluated from the central nervous system by the P-glycoprotein. Molecule 1: Piceid, Molecule 2: Emodin-6-glucoside, Molecule 3: Emodin, Molecule 4: Galloylglucose.

**Photo 1 pharmaceuticals-18-00408-p001:**
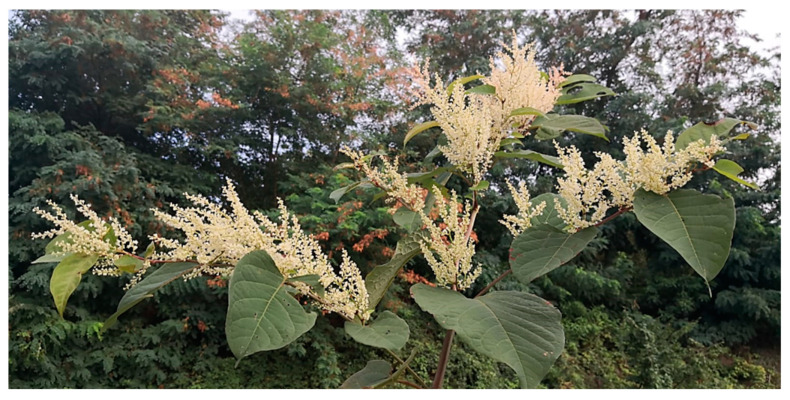
*Reynoutria japonica* Houtt. (Photo taken by Prof. Dr. Osman Tugay).

**Table 1 pharmaceuticals-18-00408-t001:** Cholinesterase inhibitory activities of *R. japonica* extracts.

Samples	Cholinesterase Inhibition (Inhibition% ± S.D. ^a^) at 100 µg/mL ^b^	
	AChE	BChE
RJF	60.89 ± 4.12 **** (IC_50_ = 64.44 ± 4.69 µg/mL)	41.13 ± 1.61 ****
RJS	64.24 ± 3.09 ^c^ **** (IC_50_ = 25.06 ± 0.52 µg/mL)	31.86 ± 4.17 **** (IC_50_ = 159.78 ± 3.50 µg/mL)
RJL	63.57 ± 2.50 **** (IC_50_ = 53.31 ± 3.38 µg/mL)	20.81 ± 3.38 **** (IC_50_ = 202.50 ± 1.41 µg/mL)
RJR	49.93 ± 4.18 **** (IC_50_ = 120.10 ± 0.14 µg/mL)	20.15 ± 1.77 ****
Galanthamine hydrobromide ^d^	90.47 ± 0.76 (IC_50_ = 0.91 ± 0.03 µg/mL)	70.95 ± 1.51 (IC_50_ = 37.94 ± 3.17 µg/mL)

^a^ Standard deviation (n: 4), ^b^ Final concentration, ^c^ 50 µg/mL, RJF: *R. japonica* flowers, RJS: *R. japonica* stems, RJL: *R. japonica* leaves, RJR: *R. japonica* roots, ^d^ Reference drug, **** *p* < 0.0001.

**Table 2 pharmaceuticals-18-00408-t002:** TYR, elastase, and collagenase inhibitory activities of *R. japonica* extracts.

Samples	Inhibition% ± S.D. ^a^ at 333 µg/mL ^b^		
	TYR	Elastase	Collagenase
RJF	- ^c^	70.06 ± 2.12 **** (IC_50_ = 111.40 ± 1.45 µg/mL)	68.82 ± 5.57 **** (IC_50_ = 231.90 ± 4.12 µg/mL)
RJS	26.03 ± 1.68 ****	65.22 ± 3.21 **** (IC_50_ = 253.03 ± 3.05 µg/mL)	78.68 ± 6.08 (IC_50_ = 236.80 ± 7.34 µg/mL)
RJL	-	80.97 ± 1.70 **** (IC_50_ = 117.20 ± 4.84 µg/mL)	81.02 ± 6.73 (IC_50_ = 171.00 ± 6.76 µg/mL)
RJR	19.46 ± 2.37 ****	9.31 ± 1.38 ****	69.68 ± 2.23 *** (IC_50_ = 160.00 ± 6.81 µg/mL)
Reference	84.56 ± 0.27 ^d^ (IC_50_ = 0.68 ± 0.05 µg/mL)	99.65 ± 0.08 ^e^ (IC_50_ = 2.65 ± 0.36 µg/mL)	87.39 ± 2.85 ^f^ (IC_50_ = 27.95 ± 0.37 µg/mL)

^a^ Standard deviation (n: 4), ^b^ Final concentration, ^c^ No activity, ^d^
*Alpha*-kojic acid (133 µg/mL) ^e^
*N*-(Methoxysuccinyl)-Ala-Ala-Pro-Val-chloromethyl ketone tested at 66.67 µg/mL ^f^ 1,10-Phenanthroline tested at 66.67 µg/mL, *** *p* < 0.001, **** *p* < 0.0001.

**Table 3 pharmaceuticals-18-00408-t003:** Compounds detected in *R. japonica* extracts.

Chemical Name	Structure	R_t_ (min)	Formula	M-H Ion	Product Ions
Galloyl glucose isomer	Phenolic acid	1.27	C_13_H_16_O_10_	331.0666	211.0251, 169.0147
Chlorogenic acid isomer	Polyphenol	3.4	C_16_H_17_O_9_	331.0863	191.0546, 135.0442
Procyanidin B1	Polyphenol	4	C_30_H_26_O_12_	577.134	559.1228, 289.0718
Catechin	Flavonoid	4.43	C_15_H_14_O_6_	289.0725	245.0827, 109.0298
Procyanidin B1	Polyphenol	5.04	C_30_H_26_O_12_	577.1346	425.0864, 289.0718
Epicatechin	Flavonoid	5.39	C_15_H_14_O_6_	289.0726	245.0833, 109.0306
Galloylprocyanidin B1/B2	Polyphenol	5.94	C_37_H_30_O_16_	729.1458	289.0712
Catechin/epicatechin gallate	Flavonoid	6.54	C_22_H_18_O_10_	441.0825	289.0716, 169.014
Piceid	Stilbenoid glucoside	6.94	C_20_H_21_O_8_	389.1221	227.0718
Catechin/Epicatechin gallate	Flavonoid	7.04	C_22_H_18_O_10_	441.082	289.0710, 169.0147
Torachrysone 8-glucoside	Naphthalene glucoside	9.85	C_20_H_23_O_9_	407.1335	269.0451, 245.0802
Emodin-6-*O*-glucoside	Anthraquinone glucoside	9.9	C_21_H_19_O_10_	431.0974	269.0462
Malonylgenistin	Flavonoid	10.46	C_24_H_21_O_13_	517.1026	473.1056, 269.0467
Torachrysone glucoside isomer	Naphthalene glucoside	10.56	C_22_H_25_O_10_	449.144	245.0804
Emodin	Anthraquinone	15.2	C_15_H_9_O_5_	269.0461	225.0570, 197.0602

**Table 4 pharmaceuticals-18-00408-t004:** Instrumental parameters of LC-MS Q-TOF analysis.

Column	Agilent Poroshell SB C-18 (3.0 mm × 100 mm × 2.7 µm)
Column temperature	35 °C
Injection volume	10 µL
Run time	32 min
Mobile phase A	0.1% formic acid in water
Mobile phase B	Acetonitrile
Flow rate	0.6 mL/min
Gradient (time-B%)	0 min-5% B
	2 min-5% B
	6 min-20% B
	18 min-70% B
	20 min-90% B
	26 min-90% B
	26.1 min-5% B
Ionization mode	Negative ESI
Drying gas temperature	325 °C
Drying gas flow	11 L/min
Nebulizer	35 psi
Capillary voltage	3000 V
Fragmentor voltage	150 V
Mass range	30–1700 amu
Reference ions	112.98587, 1033.988109

## Data Availability

Data is contained in the paper.

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
