# Peer review of "Chromatographic Analysis and Enzyme Inhibition Potential of Reynoutria japonica Houtt.: Computational Docking, ADME, Pharmacokinetic, and Toxicokinetic Analyses of the Major Compounds"

_pharmaceuticals, 2025, doi:10.3390/ph18030408_

Round 1

Reviewer 1 Report

Comments and Suggestions for Authors

The topic of this manuscript is relevant and the overall presentation is, let's say, satisfactory, but with the room for further improvements, as listed below:

  1. I would suggest combining sections 2.4 and 2.6 - there is no point in keeping 2 separate sections on drug-likeness;
  2. Lipinski's criteria of drug-likeness (both the original ones, which are ca. 25 years old, and those more recently modified) are just crude filters with many exceptions - please discuss briefly and include some more recent references.
  3. Molecular docking - please describe it in more details - software and docking conditions. I would also recommend calculating binding affinities for some appropriate ligands known for their strong binding to studied enzymes - as a positive control.
  4. Section 4.6 - upon reading it one may conclude that the molecular descriptors were calculated by the Authors themselves using the cited methods rather than returned by the SwissADME platform - please clarify.
  5. Chromatography - the Authors presented the chromatograms obtained for different parts of R. japonica plants and identified the main constituents, but in my opinion this is not sufficient to call it "fingerprinting" - is there any proof that these chromatograms are so distinctive for this plant that its identification based on chromatographic results is possible?
  6. Table 2 - a comma instead of a full stop, please correct.

Author Response

Dear Reviewer,

Thank you very much for taking the time to review this manuscript. Please find the detailed responses below and the corresponding revisions in track changes in the re-submitted files.

1. Point-by-point response to Comments and Suggestions for Authors

Comments 1: I would suggest combining sections 2.4 and 2.6 - there is no point in keeping 2 separate sections on drug-likeness;

Response 1: Thank you for pointing this out. We have combined sections 2.4 and 2.6 into a single comprehensive section titled "Drug-Likeness Parameters and ADME Profile Analysis" to improve the flow and coherence of the manuscript. This reorganization provides a more integrated discussion of the compounds' pharmacokinetic and drug-likeness properties.

Comments 2.   Lipinski's criteria of drug-likeness (both the original ones, which are ca. 25 years old, and those more recently modified) are just crude filters with many exceptions - please discuss briefly and include some more recent references.

Response 2: We appreciate this important point. We have expanded our discussion of drug-likeness beyond the traditional Lipinski's Rule of Five to acknowledge its limitations and the numerous exceptions discovered in recent years, particularly for natural products. We have added text that discusses the evolving understanding of drug-likeness parameters and cited more recent references that highlight these developments. The revised section now presents a more nuanced interpretation of the drug-likeness results.

Comments 3.   Molecular docking - please describe it in more details - software and docking conditions. I would also recommend calculating binding affinities for some appropriate ligands known for their strong binding to studied enzymes - as a positive control.

Response 3: I have included further details in the manuscript on the methods used for the docking simulations. Thank you for your suggestion. The reason we did not include a positive control is that force field-based methods do not account for all the factors that contribute to strong inhibitor binding. These methods rely on certain assumptions, which can sometimes lead to higher docking scores for experimentally validated inhibitors. This does not necessarily mean they bind better but rather reflects the limitations of the computational approach.

Comments 4.   Section 4.6 - upon reading it one may conclude that the molecular descriptors were calculated by the Authors themselves using the cited methods rather than returned by the SwissADME platform - please clarify.

Response 4: We have revised Section 4.6 to clearly state that all calculations were performed using the SwissADME web tool, which implements the various algorithms and methods cited. The revised text explicitly mentions that we utilized the platform's integrated implementation of these methods rather than performing the calculations ourselves, thus avoiding any potential misunderstanding.

Comments 5. Chromatography - the Authors presented the chromatograms obtained for different parts of R. japonica plants and identified the main constituents, but in my opinion this is not sufficient to call it "fingerprinting" - is there any proof that these chromatograms are so distinctive for this plant that its identification based on chromatographic results is possible?

Response 5: We have changed the “fingerprinting” expression to just chromatographic analyses.

Comments 6: Table 2 - a comma instead of a full stop, please correct.

Response 6: I have revised the full stops with commas in Table 2.

Reviewer 2 Report

Comments and Suggestions for Authors

1. The introduction provides a strong background on Reynoutria japonica and its phytochemistry, but it lacks a clear research gap. The authors should explicitly state: Why this study is necessary? How does it differ from previous studies? What are the novel contributions? The last paragraph of the introduction should clearly state the research objective and hypotheses.
2. The methodology lacks specific details regarding: How many replicates were used for enzyme inhibition assays? What statistical tests were applied to validate the enzyme inhibition data? Were any reference standards used to confirm the peak identities? The specific parameters used for molecular docking (e.g., grid dimensions, scoring functions) should be mentioned.
3. The enzyme inhibition results need statistical significance markers (e.g., p-values or confidence intervals) to support their claims.The LC-MS QTOF phytochemical profiling should be accompanied by a table summarizing: The detected compounds, their retention times, molecular weights, structural classifications etc. The docking results should include binding energy values in a table format for easier comparison.
4. The discussion mostly reiterates the results but lacks a comparison with previous studies on Reynoutria japonica or related species.A critical interpretation of why certain compounds exhibit stronger enzyme inhibition. A detailed explanation of toxicokinetic predictions. How do the ADME results influence the potential therapeutic applications? Also consider adding limitations of the study (e.g., lack of in vivo validation).
5. Some figures (e.g., enzyme inhibition plots, docking interaction images) could be improved with clearer axis labels and higher resolution.
6. Tables summarizing LC-MS QTOF data and docking results should be included.
7. The conclusion is well-written but could be strengthened by adding a future direction: Will there be further in vitro or in vivo studies? Highlighting the potential translational impact of the findings.

Author Response

Dear Reviewer,

Thank you very much for taking the time to review this manuscript. Please find the detailed responses below and the corresponding revisions in track changes in the re-submitted files.

1. Point-by-point response to Comments and Suggestions for Authors

Comments 1: The introduction provides a strong background on Reynoutria japonica and its phytochemistry, but it lacks a clear research gap. The authors should explicitly state: Why this study is necessary? How does it differ from previous studies? What are the novel contributions? The last paragraph of the introduction should clearly state the research objective and hypotheses.

Response 1: Thank you for pointing this out. We added a paragraph to the Introduction section.

Comments 2: The methodology lacks specific details regarding: How many replicates were used for enzyme inhibition assays? What statistical tests were applied to validate the enzyme inhibition data? Were any reference standards used to confirm the peak identities? The specific parameters used for molecular docking (e.g., grid dimensions, scoring functions) should be mentioned.

Response 2: Enzyme inhibition methods were performed with four replicates. The number of replicates (n=4) is stated as a footnote in the table in the results section. The active sites of the receptors, used as docking boxes, were identified based on the co-crystallised ligand in the crystal structures. The grid and docking parameters for the IFD simulation were set to the program's default settings.

Comments 3: The enzyme inhibition results need statistical significance markers (e.g., p-values or confidence intervals) to support their claims. The LC-MS QTOF phytochemical profiling should be accompanied by a table summarizing: The detected compounds, their retention times, molecular weights, structural classifications etc. The docking results should include binding energy values in a table format for easier comparison.

Response 3: Thank you. We performed statistical analyses and revised enzyme inhibition and the phytochemical profile tables. We also added files to the supplementary material. We have provided all the docking scores obtained using the IFD method as supplementary information. Rather than focusing solely on the top docking scores, we have considered the distribution of binding energies across various ligand poses. This approach ensures a more statistically robust analysis, as relying only on the highest scores may not always provide meaningful insights.

Comments 4:   The discussion mostly reiterates the results but lacks a comparison with previous studies on Reynoutria japonica or related species. A critical interpretation of why certain compounds exhibit stronger enzyme inhibition. A detailed explanation of toxicokinetic predictions. How do the ADME results influence the potential therapeutic applications? Also consider adding limitations of the study (e.g., lack of in vivo validation).

Response 4: Thank you for this valuable suggestion. We have discussed our results with previous studies related to the Reynoutria genus. We have expanded our discussion of toxicokinetic predictions in both Section 4.6 and the Discussion section. We have added a detailed explanation of the PAINS alerts and their implications for potential toxicity and biological specificity. We have also explicitly addressed how the ADME results directly influence potential therapeutic applications, suggesting different development strategies based on the observed pharmacokinetic profiles (e.g., topical applications for compounds with limited oral bioavailability but good enzyme inhibition). Additionally, we have acknowledged the limitation of relying on computational predictions without in vivo validation and highlighted the need for future experimental studies to confirm our findings.

Comments 5: Some figures (e.g., enzyme inhibition plots, docking interaction images) could be improved with clearer axis labels and higher resolution.

Response 5: We have changed the figures with the higher resolution ones.

Comment 6: Tables summarizing LC-MS QTOF data and docking results should be included.

Response 6: We have included details in Supplementary Material files.

Comment 7: The conclusion is well-written but could be strengthened by adding a future direction: Will there be further in vitro or in vivo studies? Highlighting the potential translational impact of the findings.

Response 7: Thank you. We have added a future direction and suggestion also to the conclusion section.

Round 2

Reviewer 1 Report

Comments and Suggestions for Authors

I am satisfied with the corrections - in my opinion the manuscript has improved significantly. There are, however, some minor issues to be addressed, e.g.

1. The last paragraph of Section 3:

"Our computational toxicokinetic analyses revealed several structural alerts that may impact the safety and specificity of these compounds' biological activities. Emodin and emodin-6-glucoside triggered quinone_A alerts, indicating potential redox cycling that could generate reactive oxygen species and resulting in non-specific cytotoxicity. Galloylglucose was flagged for containing a catechol moiety, which may chelate metal ions and inhibit various enzymes non-selectively. Piceid's stilbene group, while central to its bioactivity, may cause aggregation or auto-fluorescence in certain assay conditions. These findings have important implications for therapeutic development, as they suggest potential off-target effects and toxicity risks that should be addressed through structural modifications or targeted delivery strategies. Moreover, the predicted ADME properties directly influence potential therapeutic applications: compounds with poor oral bioavailability but favorable enzyme inhibition profiles (like the glycosylated derivatives) may be better suited for topical formulations targeting skin aging, while compounds with BBB permeability (such as emodin) might have greater potential for neurological applications if their toxicity concerns can be mitigated. A key limitation of our study is the reliance on computational predictions without in vivo validation, highlighting the need for future pharmacokinetic studies to confirm these findings before advancing to clinical applications."

seems to be a slightly expanded repetition of some statements that are given in the same section, but a page earlier - please make sure that there are no unnecessary repetitions in the text

2. The BOILED-egg figure is of poor quality, the egg white is invisible. I know it is generated automatically, but could the Authors please edit it in Paint or any similar software and draw a line around the egg white so that we know which compounds are expected to be absorbed from the GI tract and which are not.

3. When I opened the revised manuscript PDF file in Acrobat, the BOILED egg figure overlapped with the Figure 6 caption - please make sure that there are no formatting issues.

Author Response

Dear Editor and Reviewers,

We thank the reviewers for their constructive feedback and suggestions to improve our manuscript. We have carefully addressed all the points raised:

Response to Reviewer 1:

  1. Regarding the redundancy in Section 3: We have removed the redundant paragraph at the end of Section 3 to eliminate repetition and improve the flow of the manuscript.

(This change might not appear in the track changes section of the document because the deleted part was also included to the document in the first set of reviewer changes.)

  1. Regarding the BOILED-Egg figure quality: We have modified Figure 7 as requested. While the figure is automatically generated by computational software, which limits our ability to add perfect thin outline borders, we have addressed the visibility concern by enhancing the contrast between regions and adjusting the background color. The modified figure now shows clearer boundaries between the egg white (representing GI absorption) and both the background and the yellow yolk (representing BBB permeation). These color adjustments make it much easier to distinguish which compounds are predicted to be absorbed from the GI tract versus those that can penetrate the blood-brain barrier.
  2. Regarding the formatting issue: We have fixed the formatting problem where the BOILED-Egg figure overlapped with the Figure 6 caption. We have adjusted the spacing between figures and ensured that all figure captions are properly aligned with no overlaps.

(The issue likely stems from the journal system automation. We will continue to monitor for any problems in the upcoming galley proofs.)

Reviewer 2 Report

Comments and Suggestions for Authors

This revised response improved clarity, provides more details on statistical methods, and explicitly states how concerns were addressed.

Author Response

Dear Editor and Reviewers,

We thank the reviewers for their constructive feedback and suggestions to improve our manuscript. We have carefully addressed all the points raised:

Response to Reviewer 2:

We thank the reviewer for acknowledging the improvements in our revised manuscript regarding clarity, statistical methods details, and our explicit responses to previous concerns.

We believe these modifications have further improved the clarity and presentation of our manuscript.

Sincerely,
